# Curriculum by Smoothing

**Samarth Sinha [1], Animesh Garg [1, 2], Hugo Larochelle [3]**

## Abstract

Convolutional Neural Networks (CNNs) have shown impressive performance in computer vision tasks such as image classification, detection, and segmentation. Moreover, recent work in Generative Adversarial Networks (GANs) has highlighted the importance of learning by progressively increasing the difficulty of a learning task [26]. When learning a network from scratch, the information propagated within the network during the earlier stages of training can contain distortion artifacts due to noise which can be detremental to training. In this paper, we propose an elegant curriculum-based scheme that smoothes the feature embedding of a CNN using anti-aliasing or low-pass filters. We propose to augment the training of CNNs by controlling the amount of high frequency information propagated within the CNNs as training progresses, by convolving the output of a CNN feature map of each layer with a Gaussian kernel. By decreasing the variance of the Gaussian kernel, we gradually increase the amount of high-frequency information available within the network for inference. As the amount of information in the feature maps increases during training, the network is able to progressively learn better representations of the data. Our proposed augmented training scheme significantly improves the performance of CNNs on various vision tasks without either adding additional trainable parameters or an auxiliary regularization objective. The generality of our method is demonstrated through empirical performance gains in CNN architectures across four different tasks: transfer learning, cross-task transfer learning, and generative models. The code will soon be released at www.github.com/pairlab/CBS.

## 1 Introduction

Deep Learning models have revolutionized the field of computer vision, which has led to great progress in recent years. Convolutional Neural Networks (CNNs) [33], have turned out to be a very effective class of models, which have enabled state-of-the-art performance on a multitude of computer vision tasks such as image recognition [31, 19], semantic segmentation [38, 46], object detection [13, 45], pose estimation [62] to name a few.

Recent work by Karras et al. [26] showed excellent results on building a curricula to progressively increase the learning task for a GAN. By simply increasing the resolution of the image progressively, they are able to achieve significantly better results and stabilize GAN training. This progressive curricula helps the CNN models learn and generate better representations during GAN training. However a core question remains: *how to design a curricula that fundamentally improves the ability of CNNs to learn better representations from data?*

In this paper, we propose an elegant and effective curriculum that augments a CNN's training regime by smoothing the feature maps of a CNN using low-pass or anti-aliasing filters, and progressively adding the high-frequency information in the feature maps to the model. Specifically, we propose to learn CNNs using a curricula, such that the high-frequency information can only be used by feature maps towards the later stages of training. As shown by [25], early stages of training is

Corresponding author: samarth.sinha@mail.utoronto.ca

critical in learning for deep networks. The proposed method improves training during early stages by reducing the noise propagated by the untrained parameters in the feature space by convolving the output of each CNN layer by Gaussian filters. During the early stages of training, a network propagates a significant amount of noise due to the untrained parameters, therefore by using a low-pass or specifically a Gaussian filter, we are able to smooth the noise and reduce any aliasing artifact in the feature space. Furthermore, as the network parameters converge to the optimal solution, and the noise in the feature maps decreases, we anneal the standard deviation of the Gaussian filters which therefore increases the information propagated within the network, and allowing the network to learn richer representations from the newly available high-frequency information.

The Gaussian kernel is known as a low-pass filter with anti-aliasing properties, which smooths high-frequency information from the input. For a Gaussian kernel, the variance parameter controls the amount of high-frequency information that will be filtered; therefore, by annealing the standard deviation of the Gaussian kernels, we can intuitively control the amount of high-frequency information within the layers over time and consequently improve the performance of deep networks on downstream vision tasks. It is also worth noting that the proposed method also adds no additional trainable parameters, is generic, and can be used with any CNN-variant.

The main contributions of this paper can be summarized as:

- We introduce an elegant and effective curricula that utilizes feature smoothing in CNNs to reduce the amount of noise, due to untrained parameters, in the feature maps. This information is progressively added which leads to improvement in learned feature maps in CNNs.
- We conduct image classification experiments using commnly-used vision datasets and CNN architecture variants to evaluate the effect of controlling smoothing feature maps during training.
- We evaluate the models trained on ImageNet, with and without our proposed curricula, as feature extractors to train "weak" classifiers on previously unseen data. We also use the pretrained CNNs on different vision tasks where ImageNet pretraining is important: semantic segmentation and object detection, and significantly outperform models trained with curricula.
- Finally, we show further improvements on representation learning using VAEs [28] and on Zero-shot Domain Adaptation to highlight the generality of the solution as well as the robustness of the learned representations.

## 2 Background and Preliminaries

Given a labeled dataset of the form $(x_i, y_i)_{i=1}^N$, $y_i \in Y$ represents the ground-truth label for the input image $x_i \in X$. For a given dataset, the network is optimized by

$$\min_\theta \frac{1}{N} \sum_{i=1}^N \mathcal{L}_T(\theta(x_i), y_i)$$

where $\mathcal{L}_T$ represents the task-specific, differentiable loss function and $\theta$ is a parameterized neural network. Since our proposed method is a general modification to learning in CNNs, any task-specific $\mathcal{L}_T$ can be used for training.

### 2.1 Convolutional Neural Networks

To denote the convolutional operation of some kernel $\theta_k$ on some input $h_i$, we will use $\theta_k \circledast h_i$ In deep learning, a typical CNN is composed of stacked trainable convolutional layers [33], pooling layers [6], and non-linearities [41]. A typical CNN layer can be mathematically represented as

$$h_i = ReLU(pool(\theta_w \circledast x_i)) \tag{2.1}$$

where $\theta_w$ are the learned weights of the convolutional kernel, $pool$ represents a pooling layer, $ReLU$ is an example of a non-linearity [41] and $h_i$ is the output of the hidden layer.

### 2.2 Gaussian Kernels

Gaussian kernels are deterministic functions of the size of the kernels and standard deviation $\sigma$. A 2d Gaussian kernel can be constructed using:

$$k(x, y) = \frac{1}{2\pi\sigma^2} \exp\left(-\frac{x^2 + y^2}{2\sigma^2}\right)$$

where $k(x, y)$ represent the $x$ and $y$ spatial dimensions in the kernel.

Gaussian kernels have been extensively studied and used in traditional image processing and computer vision, such as Scale Space Theory [36, 35, 53, 10]. Scale space theory aims to increase the scale-invariance of traditional computer vision algorithms by convolving the image with a Gaussian kernel. Scale space theory has been applied to corner detection [68], optical flow [1], modeling multi-scale landscapes [5], and more recently to CNNs [67]. Gaussian kernels have also been widely used as low-pass filters in signal processing [64, 50, 8]. Recent work has also used fixed Gaussian kernels for their anti-aliasing properties in deep learning [34, 39, 4].

### 2.3 CNN Training Improvements

Since CNNs were proposed originally by [33], there have been many significant improvements proposed to stabilize training, and improve the expressiveness of these networks. Recently, deep CNNs have been popularized by [31], and many deep architectural variants have since been popularized [19, 51, 56, 57]. There have also been different normalization methods that have been proposed to increase the network generalization [54, 49] and training of the models [24, 2, 60, 61, 23].

### 2.4 Curriculum Learning

Curriculum learning was originally defined by [3] as a way to train networks by organizing the order in which tasks are learned and incrementally increasing the difficulty of a task. Curriculum learning has been a popular area of study for reinforcement learning agents [12, 55, 40]. Recent work has also proposed to use curriculum learning for RNNs [16, 66]. More recent work in curriculum learning in deep learning considers learning an SVM to measure the difficulty of a task performed [52], introducing sample-wise differentiable data parameters to govern how to sample [48], and methods that learn a curricula for a specific domain, such as Information Retrieval [43].

### 2.5 Pre-trained CNNs

Pre-trained CNNs have been thoroughly explored in transfer learning [22, 29], and are essential for performance on various vision tasks [17, 38, 13, 45]. Typically, networks are pre-trained on a large-scale image classification dataset (commonly the ImageNet dataset [47]), and transferred to a different downstream task such as semantic segmentation or object detection. Since many classical vision tasks rely on pretrained networks, it is important to learn better representations during pretraining as recent work suggests that networks that perform better on the pretrained task tend to transfer better [44].

## 3 Curriculum By Smoothing

### 3.1 Gaussian Kernel Layer

Similar to a kernel in a convolutional layer, Gaussian kernels are a parameterized kernel with a standard deviation given by $\sigma$. The $\sigma$ hyperparameter of the kernel controls how much of the output will be "blurred" after a convolution operation, as increasing $\sigma$ results in a greater amount of blur. An alternate interpretation of a Gaussian kernel is as a low-pass filter, which masks high-frequency information from the input, depending on the choice of $\sigma$. By adding "blur", we remove high frequency information from the input. Unlike the kernels of a CNN, Gaussian kernels are not trained via backpropagation, and are deterministic functions of $\sigma$.

We propose to augment a given CNN using a Gaussian Kernel layers. The Gaussian Kernel layer is proposed to be added to the output of each convolutional layer in a CNN to minimize noise added by the untrained parameters early in training. Formally, this can simply be added to Eqn. 2.1 as

$$h_i = ReLU(pool(\theta_{G_\sigma} \circledast (\theta_w \circledast x_i))) \tag{3.1}$$

where the $\theta_{G_\sigma}$ is a Gaussian kernel with a chosen standard deviation $\sigma$.

By applying the Gaussian blur to the output of a convolutional layer, we smooth out features of the CNN outputs, and reduce high-frequency information in the CNN. We perform this operation on the output of each CNN layer, to smooth all the feature maps in the CNN. After the smoothing

operation, the network still has low-frequency information, to optimize the task-objective, $\mathcal{L}_T$. By reducing the noise and aliasing artifacts in the feature maps, we ensure that the are not biased by the noise in the weights initializations, as such bias early in training has been shown to be critical for learning [25].

## 3.2 Designing a Curriculum

To design an effective curricula for training CNNs, we aim to progressively add information to allow for the networks to progressively adapt as more information is propagated in the networks. We propose to bias the training of CNNs by first focusing on low-frequency information using a high value of $\sigma$ for all $\theta_{G_\sigma}$ in the network. By annealing the value of $\sigma$ as training progresses, the network naturally learns from the increased availability of information in the feature maps. From the perspective of signal processing, a Gaussian kernel has anti-aliasing properties [67], therefore using a high value of $\sigma$ during early stages can also be viewed as high anti-aliasing on the feature maps. The feature maps produced by untrained parameters contain high amount of aliasing information the network has not learned a good representation of the data. Such information is smoothed out using a Gaussian kernel.

As we anneal $\sigma \to 0$, we recover regular training for CNNs. But the core component of the algorithm relies in the early stages of training. There we reduce the effect of the noise from untrained parameters and ensure that they do not harm training as the early stages is the most critical time for training deep networks [25]. As the networks get increasingly better at the task, we provide the network with more information, since the noise from of the parameters and aliasing artifacts will have decreased. CBS is a general method for training CNNs, which can be applied to any CNN variant. A sample PyTorch-like code snippet is available in below for a two-layer CNN, to illustrate its ease of implementation [42]. In the sample pseudo-code, gaussian_kernel, conv1 and conv2 represent convolutional operations on the input. A normalization layer such as Batch Normalization [24] may also be used like in many modern architectures [19, 65, 63, 51].

```
1 # Use the Gaussian kernel after the convolution operation
2 h = gaussian_kernel(conv1(x))
3 # Add non-linearity and pooling
4 h = activation(pool(h))
5
6 # Same operation after each conv. layer
7 h = gaussian_kernel(conv2(h))
8 h = activation(pool(h))
```

## 4 Experiments

We have designed the experiments to evaluate our proposed scheme by evaluating the following hypotheses:

- **Better task performance:** How does the performance vary when a network is trained with or without our curriculum learning method?
- **Better feature extraction:** How does the trained network perform when it is used to extract features from a different dataset to train a weak classifier and for different vision tasks where pretraining is required?
- **Generative Models:** How does adding CBS help with distinctly different vision tasks that utilize CNNs such as generative models?

A CNN trained on a large-scale dataset, such as ImageNet [47], is able to learn useful representations and semantic relationships in natural images. The pretrained network can be used to extract features from an image to make the classification task easier. A *better* CNN model should be able to extract *better* features from unseen images. Since the goal of CNNs is to learn better representations from data, it is important to evaluate the networks trained with CBS as a feature extractor. To evaluate a model on its ability as a feature extractor, we $i)$ freeze the weights of the model and train only a weak classifier on the feature outputs of a new dataset, $ii)$ pretrain on a large scale dataset, specifically ImageNet [47], and use the learned model to perform semantic segmentation and object detection,

Table 1: **Image Classification**. Top-1 classification accuracy on CIFAR10, CIFAR100 and SVHN for CNNs trained normally and CNNs trained using CBS. We show significant improvements over three standard datasets using four different network architectures: VGG-16 [51], ResNet18 [19], Wide ResNet-50 [65], and ResNext-50 [63].

|  | SVHN | CIFAR10 | CIFAR100 |
|---|---|---|---|
| VGG-16 | $96.6 \pm 0.2$ | $85.8 \pm 0.2$ | $57.0 \pm 0.2$ |
| VGG-16 + CBS | $\mathbf{97.0} \pm 0.2$ | $\mathbf{88.9} \pm 0.3$ | $\mathbf{61.4} \pm 0.3$ |
| ResNet-18 | $97.2 \pm 0.2$ | $87.1 \pm 0.3$ | $62.4 \pm 0.3$ |
| ResNet-18 + CBS | $\mathbf{98.7} \pm 0.2$ | $\mathbf{90.2} \pm 0.3$ | $\mathbf{65.4} \pm 0.2$ |
| Wide-ResNet-50 | $97.7 \pm 0.1$ | $91.8 \pm 0.1$ | $73.3 \pm 0.1$ |
| Wide-ResNet-50 + CBS | $\mathbf{98.3} \pm 0.3$ | $\mathbf{93.9} \pm 0.1$ | $\mathbf{75.9} \pm 0.2$ |
| ResNeXt-50 | $97.7 \pm 0.2$ | $93.1 \pm 0.1$ | $74.1 \pm 0.3$ |
| ResNeXt-50 + CBS | $\mathbf{99.0} \pm 0.2$ | $\mathbf{95.1} \pm 0.2$ | $\mathbf{77.0} \pm 0.1$ |

Table 2: **Image Classification**. Top-1 and Top-5 classification accuracy on ImageNet for CNNs trained normally and trained using CBS. We see significant improvements on for VGG-16 and a ResNet-18 networks when trained with CBS.

|  | ImageNet (Top-1) | ImageNet (Top-5) |
|---|---|---|
| VGG-16 | $63.45 \pm 0.4$ | $83.81 \pm 0.3$ |
| VGG-16 + CBS | $\mathbf{66.02} \pm 0.5$ | $\mathbf{86.26} \pm \mathbf{0.3}$ |
| ResNet-18 | $67.90 \pm 0.7$ | $85.86 \pm 0.5$ |
| ResNet-18 + CBS | $\mathbf{71.02} \pm 0.8$ | $\mathbf{89.55} \pm 0.6$ |

and $iii$) evaluate the models ability to learn robust representations from data and evaluate on a zero-shot domain adaptation digit recognition task (ZSDA).

We compare our method with the standard training procedure (without curriculum learning) for CNNs. Training a CNN with backpropogation is a very competitive baseline, as it is the prevalent training paradigm used [33]. In this section we will refer to a CNN trained normally as **CNN** and a CNN trained using Curriculum By Smoothing as **CBS**. Unless otherwise noted, for all experiments, except ImageNet, we use an initial $\sigma$ of 1, a $\sigma$ decay rate of 0.9, and decay $\sigma$'s value every 5 epochs. For ImageNet we decay the value of $\sigma$ two times every epoch, by the same factor, since the dataset is significantly larger in size.

**Image Classification**   For image classification we evaluate the performance of our curriculum based networks on standard vision datasets. We test our methods on CIFAR10, CIFAR100 [30] and SVHN [15]. CIFAR10 and CIFAR100 are image datasets with 50,000 samples, each with 10 and 100 classes, respectively. SVHN is a digit recognition task consisting of natural images of the 10 digits collected from "street view", and it consists of 73,257 images. Finally, to prove that our network can scale to larger datasets, we evaluate on the ImageNet dataset [47]. The ImageNet dataset is a large-scale vision dataset consisting of over 1.2 million images spanning across 1,000 different classes.

In our experiments, we work with 4 different network architectures: VGG-16 [51], ResNet-18 [19], Wide-ResNet-50 [65], and ResNeXt-50 [63]. For optimization, we use SGD with the same learning rate scheduling, momentum and weight decay as stated in the original paper, *without hyperparameter tuning*. The task objective, $\mathcal{L}_T$, for all the image classification experiments is a standard unweighted multi-class cross-entropy loss. For all experiments, except ImageNet, we report the mean accuracy over 5 different seeds. For ImageNet, we report the mean performance over 2 seeds. All experimental results for CIFAR10, CIFAR100 and SVHN are listed in Table 1 where we report the top-1 accuracy. The results for ImageNet are tabulated in Table 2, where we report the Top-1 and Top-5 classification accuracy.

Table 3: **Feature Extraction for Classification**. Top-1 classification accuracy on CIFAR10, CIFAR100 and SVHN when the CNNs trained on ImageNet normally and CNNs trained using Curriculum By Smoothing (CBS) are used as feature extractors on a different dataset. The CNN weights are then frozen, and the features from the images are used to train a 3-layer Multi-Layer Perception with ReLU activation. We show considerable improvement for all three datasets, as well as both network architectures.

|  | SVHN | CIFAR10 | CIFAR100 |
|---|---|---|---|
| VGG-16 | $69.31 \pm 0.2$ | $71.94 \pm 0.2$ | $46.10 \pm 0.1$ |
| VGG-16 + CBS | $\mathbf{72.04} \pm 0.2$ | $\mathbf{73.82} \pm 0.3$ | $\mathbf{48.79} \pm 0.1$ |
| ResNet-18 | $72.12 \pm 0.5$ | $72.98 \pm 0.5$ | $51.30 \pm 0.3$ |
| ResNet-18 + CBS | $\mathbf{76.30} \pm 0.5$ | $\mathbf{75.92} \pm 0.6$ | $\mathbf{54.99} \pm 0.3$ |

Table 4: **Transfer Learning**. Results for transfer learning on a different task on the Pascal VOC Dataset. For all semantic segmentation experiments we use Fully Convolutional Network with VGG-16 network, trained on ImageNet from Section 4. For all Object Detection experiments we use Fast-RCNN with the same VGG-16 backbone.

|  | Semantic Segmentation (% mIoU) | Object Detection (% mAP) |
|---|---|---|
| CNN | $55.7 \pm 0.2$ | $67.9 \pm 0.4$ |
| CBS | $\mathbf{57.9} \pm 0.3$ | $\mathbf{70.0} \pm 0.2$ |

We see that using our method, we are able to obtain better results across the three datasets in Table 1 in all 4 network architectures. By augmenting the normal training paradigm, we are able to learn better representations from the images, and therefore significantly improve the performance on all tasks. The fact that the results show improvement with each of the network architectures, suggests that we are able to fundamentally improve CNN training. Another noteworthy observation from the results show that as the image classification task becomes more difficult, CBS networks are able to outperform the baseline CNN by an increasing margin. Similarly, the ImageNet results in Table 2 further demonstrate how we are able to scale our method to work in large scale settings. By outperforming regular CNNs on ImageeNet and the other baseline datasets, CNNs trained using CBS can scale to large datasets, and modern CNN architectures.

**Feature Extraction**  Utilizing the VGG-16 networks trained on ImageNet from Section 4, we freeze the CNN weights, and use a 3 layer fully connected network with 500 hidden units in each layer and ReLU activations [41]. In all the experiments, the networks are trained using the Adam optimizer [27] with a learning rate of $10^{-4}$ for 20 epochs. To test the ability of the network as a feature extractor, we test the network on the CIFAR10, CIFAR100 [30] and the SVHN dataset [15]. By freezing the weights of the CNN, we ensure that the only factor influencing the performance is the ability of the CNN to extract features from a novel data distribution than what the network was originally trained on. Similary to 4, the task objective $\mathbb{L}_T$ is an unweighted cross-entropy loss. The results of the experiment are summarized in Table 3.

Similar to image classification, we see a similar boost in performance even when we simply transfer the weights. We note that the observation that better ImageNet classifiers also better transfer to an unseen dataset has previously been explored in [29], which shows that there is a strong correlation between the performance of a model trained on ImageNet and its ability to transfer when used as a feature extractor or after fine-tuning. Our contribution is to show that this improved transfer can be achieved not by changing the network's architecture (which we fix to VGG), but by adjusting the training procedure of the pretrained network. Indeed, [29] note that successful transfer is quite sensitive to the inductive bias of training and commonly used regularizers actually worsen transfer performance, despite having good performance on ImageNet.

**Transferring to Different Task**  Similar to the ability of a network to *generalize* to unseen data, a trained CNN should also be able to *adapt* to a new task. A networks ability to adapt to a different downstream task is very important in computer vision since many tasks, such as semantic segmentation and object detection, depend on pretrained large-scale classifiers (typically ImageNet) which

Table 5: **Zero-Shot Domain Adaptpation**. Comparison of different architectures trained with and without curricula for ZSDA for the digit recognition task. We present mean and standard deviation over 5 runs. Adding CBS during training improves each networks performance by learning better and more robust representations, which then improves zero-shot performance.

| Source → Target | Backbone | MNIST → USPS | USPS → MNIST | SVHN → MNIST |
|---|---|---|---|---|
| Source Only | WideResnet-50 | $79.37 \pm 0.24$ | $46.66 \pm 0.64$ | $72.70 \pm 0.18$ |
| Source Only + CBS | WideResnet-50 | $\mathbf{81.69} \pm 0.24$ | $\mathbf{49.89} \pm 0.21$ | $\mathbf{74.32} \pm 0.45$ |
| Source Only | ResNext-50 | $70.70 \pm 0.31$ | $40.74 \pm 0.24$ | $64.45 \pm 0.23$ |
| Source Only + CBS | ResNext-50 | $\mathbf{71.23} \pm 0.12$ | $\mathbf{43.35} \pm 0.20$ | $\mathbf{67.89} \pm 0.84$ |
| Target Only | WideResnet-50 | $96.29 \pm 0.21$ | $99.85 \pm 0.10$ | $99.85 \pm 0.10$ |

are then fine-tuned on the novel task. In this section we evaluate the ability of CNNs trained with and without CBS to adapt to the new task of semantic segmentation and object detection.

For semantic segmentation we use a Fully Convolutional Network (FCN-32) [38], with an ImageNet pretrained VGG-16 backbone from Section 4. For object detection we utilize a Faster-RCNN model [45], with the same pretrained VGG-16 backbone. We train each model with the same training setup as proposed in the original respective paper for the PASCAL-VOC dataset [11].

**We do not tune any hyperparameter for either set of experiments.** $\mathcal{L}_T$ is simply the pixel-wise unweighted cross-entropy loss for semantic segmentation. For object detection, $\mathcal{L}_T$ is the sum of the regression (smooth $\ell$-1) loss for bounding box prediction, and a classification (cross-entropy) loss for classifying the object in the bounding box. We report the networks for semantic segmentation using the mean Intersection over Union (mIoU) and for object detection using mean Average Precision (mAP). The results for both segmentation and detection are in Table 4.

We see that training the networks using CBS outperforms regular CNNs by a good margin for both tasks. The improvement in scores further suggests that CBS improves the training regime for CNNs, and makes them better at feature extraction. The pretrained ImageNet models trained with CBS are not just superior at performing ImageNet classification, but also fundamentally better as feature extractors. CBS improves the critical early stages of training which then results in significantly improved downstream performance.

**Zero-shot Domain Adaptation**  To further evaluate on learning robust representations, we test the model on the task of zero-shot domain adaptation. We evaluate the model on the standard zero-shot digit recognition task as in [59], using two different network architectures: Wide-ResNet-50 [65], and ResNeXt-50 [63]. The results for zero-shot domain adaptation (ZSDA) are summarized in 5. We train the models on a given source dataset, and then evaluate on a novel target dataset. We see that simply adding CBS to each network architecture, we are able to significantly improve the performance of the network by learning better, more robust representations from the source data. This shows that by adding a curricula, we are able to learn better generalizable representations from the source data, that can transfer better to a novel target distribution.

**Generative Models**  To test the generality of our proposed solution we consider the task of generative models, and use Variational AutoEncoders [28], to evaluate learning unsupervised representations from data. We consider two popular variants: VAE [28] and the $\beta$-VAE ($\beta = 10$) [20]. Our results are summarized in Table 6 where we report the NLL, Mutual Information and the number of Active Units for the MNIST and CelebA benchmark datasets [37]. The datasets considered are inherently very different since MNIST is a binary dataset consisting of handwritten digits, and CelebA models natural images of faces. We use the same network architecture as [58] and use a 50-dimensional latent space. We add CBS to each convolutional and transpose-convolutional layer of the network. We describe the metrics and how to compute them in the Appendix A.

By significantly improving the NLL of the baseline VAE, we improve the ability of the network to learn *better* reconstructions from images. Furthermore, Mutual Information (MI) and the number of active units evaluate the learned latent of the VAE. Improving upon both metrics on both datasets shows that along with learning better reconstructions, we also learn a richer posterior. By showing significant improvements for both datasets, we show that CBS can also be useful for generative

Table 6: **Unsupervised Representation Learning with Generative Models**. We evaluate the ability of a model to learn unsupervised representations from data using a VAE [28] and a $\beta$-VAE [20], with $\beta = 10$ on two benchmark representation learning datasets: MNIST [32] and CelebA [37]. We show that using CBS, we are able to learn significantly better reconstructions, as shown by NLL, and richer latent spaces, as shown by Mutual Information and # of Active Units. We report the mean and standard deviation over 3 random seeds.

| MNIST [32] | NLL | Mutual Information | # of Active Units |
|---|---|---|---|
| VAE | $83.9 \pm 0.2$ | $125.0 \pm 0.8$ | $\mathbf{36} \pm 0.5$ |
| VAE + CBS | $\mathbf{82.0} \pm 0.3$ | $\mathbf{127.3} \pm 0.4$ | $\mathbf{36} \pm 0.3$ |
| $\beta$-VAE | $126.1 \pm 0.5$ | $6.3 \pm 0.3$ | $8 \pm 1.1$ |
| $\beta$-VAE + CBS | $\mathbf{125.0} \pm 0.2$ | $\mathbf{7.2} \pm 0.4$ | $\mathbf{11} \pm 0.9$ |
| CelebA [37] | NLL | Mutual Information | # of Active Units |
| VAE | $66.1 \pm 0.4$ | $108.5 \pm 0.8$ | $44 \pm 0.9$ |
| VAE + CBS | $\mathbf{64.9} \pm 0.3$ | $\mathbf{108.7} \pm 0.6$ | $\mathbf{48} \pm 0$ |
| $\beta$-VAE | $92.6 \pm 0.3$ | $3.6 \pm 0.1$ | $34 \pm 0.9$ |
| $\beta$-VAE + CBS | $\mathbf{91.3} \pm 0.3$ | $\mathbf{3.9} \pm 0.3$ | $\mathbf{34} \pm 0.5$ |

Table 7: **Ablation study**. Applying smoothing to different components of the network. We report the mean and standard deviation over 5 random seeds using a ResNet-18. We see that applying Gaussian smoothing on the images or without decaying the value of $\sigma$, the network is unable to learn effective representations.

| | Image Only | Image + Features | Constant $\sigma = 1$ | Network | CBS |
|---|---|---|---|---|---|
| CIFAR-10 | $80.0 \pm 0.3$ | $84.1 \pm 0.2$ | $85.3 \pm 0.4$ | $87.1 \pm 0.3$ | $\mathbf{90.2} \pm 0.3$ |
| CIFAR-100 | $45.7 \pm 0.3$ | $49.6 \pm 0.3$ | $54.0 \pm 0.2$ | $62.4 \pm 0.3$ | $\mathbf{65.4} \pm 0.2$ |

models, regardless of the target distribution. Furthermore, showing improvements on unsupervised representation learning, along with supervised learning in previous sections, we show that CBS is a fundamental tool for learning better CNNs.

**Ablation Study**   In this section we investigate the reason for why CBS works. We analyze the effect of adding CBS directly onto the image, the image and the layers, and also using a constant value of $\sigma$. The results in Table 7 show that using the low-pass filter directly on the images significantly hurts the performance of the model with or without CBS applied to the rest of the network. Applying CBS to the images likely takes away meaningful information that is essential to training. Since CBS smooths the feature maps against aliasing artifacts caused by the parameters, it supports that adding adding CBS to the images, similar to [26], will not help. Similarly, without progressively allowing more information to the network, and instead using a constant value for $\sigma$, the network does not perform as well as the baseline architecture. Both the results show confirm the two components of CBS: $i$) performing anti-aliasing directly on the learned features and $ii$) the importance of annealing $\sigma$. We perform further experiments in Appendix B where we investigate the effect of adding CBS on single layers, and in Appendix C where we discuss the effect of different values of $\sigma$ and decay rate. We finally investigate the choice of the initialization scheme on the performance of the model in Appendix D, where we discuss how CBS is more robust to the choice of initialization.

## 5   Conclusion

In this paper we describe a simple yet effective curricula for training CNNs, using a Gaussian kernel. During training, we propose to convolve the output of a CNN using such a Gaussian kernel to smooth out the feature map, and reduce the aliasing caused by the untrained network parameters. As the network training, we progressively anneal $\sigma$, and allow more high-frequency information to be propagated within the network. Using our technique, Curriculum By Smoothing, we are able to learn CNNs that $i$) perform better on the task of image classification, $ii$) perform better generalization when used as feature extractors on unseen datasets by learning more robust representations from data, $iii$) use the pretrained weights for other vision tasks that require pretraining such as object detection and semantic segmentation and $iv$) improve VAEs for unsupervised representation learning

on benchmark datasets. Future extensions to the work can look to combine CNNs with traditional signal processing, and strengthen the connections between the two fields.

## Broader Impact

In this paper we describe a technique to fundamentally improve training for CNNs. This paper has impact wherever CNNs are used, since they can also be trained using the same regime, which would results in improved task-performance. Applications of CNNs, such as object recognition, can be used for good or malicious purposes. Any user or practitioner has the ultimate impact and authority on how to deploy such a network in practice. The user can use our proposed strategy to improve their underlying machine learning algorithm, and deploy it in whichever way they choose.

## Acknowledgements

We would like to thank Anirudh Goyal for insightful discussions and helpful feedback on the draft. We would also like to thank Jiajun Wu for insightful initial discussions. We acknowledge the funding from the Canada CIFAR AI Chairs program. Finally, we would like to acknowledge Nvidia for donating DGX-1, and Vector Institute for providing resources for this research.

## Footnotes

[1] University of Toronto, Vector Institute, [2] Nvidia, [2] Mila, Google Brain, CIFAR Fellow

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
