[Supplementary Material]

## A  VAE Experiments

The hyperparameters for these experiments are discussed in C. In all experiments, we use the same VAE-architecture as [58], and reshape the images to be $32 \times 32$. The size of the latent dimension is set to be 50 therefore the maximum # of Active Units is 50. The metrics considered to measure the performance of VAEs include Negative Log-Likelihood, # of Active Units in the latent space and the Mutual Information between the input $x$ and the latent space $z$, $I(z; x)$. We use the same formulation as [7] to compute NLL and # of Active Units, and use the same formulation as [9, 21] to approximate the Mutual Information. We train each VAE for 100 epochs using Adam optimizer [27] with a fixed learning rate of $1^{-4}$ and a mini-batch size of 64.

## B  CBS on Single Layer

Table 8: Ablation study by applying a Gaussian kernel on only specific layers of a simple 3 layer CNN with Max-Pooling and ReLU activations. **Bolded** value corresponds to the proposed model which uses the kernel layer after each convolutional layer.

| None | First | Second | Third | All |
|---|---|---|---|---|
| 57.04 | 60.03 | 59.87 | 59.88 | **61.41** |

Another important factor to investigate is to look at how applying a Gaussian kernel to each layer changes the performance of the CNN. For this experiment we use a simple 3-layer CNN, with Max-Pooling and ReLU activations. We apply the kernel to each layer individually and evaluate its performance on the CIFAR10 dataset. For comparison, we report a network with no Gaussian kernel layers, and a network with Gaussian kernel layers after each convolutional layer (proposed model). The results are presented in Table 8. Clearly, the best performing model is the proposed model: when one applies Gaussian kernels after each layer. But just by applying the kernel to any layer, there is a notable boost in performance over the baseline. By adding the kernel to only one layer, the network as a whole does not progressively learn, which is important

## C  Hyperparameter Discussions

The main important hyperparamters to consider for CBS are the initial value of $\sigma$ and the decay rate applied. In practice, the ideal choice of value for $\sigma$ and the decay rate depended solely on the speed of convergence for the model and the size of the model. For all Resnet variants (ResNet [19], ResNeXt [63] and Wide-ResNet [65]) we use $\sigma = 1$ and a decay rate of 0.9. Since VGG-16 has significantly more parameters, and [19] show that it is slower in convergence, a higher value of $\sigma = 2$ was used. Since we know that a VGG network takes more epochs till it converges and therefore remains further from the optimal parameters for more epochs, more smoothing is required to minimize the anti-aliasing artifacts during training. This supports the hypothesis that more smoothing is required when the weights are far from convergence.

The same values were used for all experiments utilizing the architecture. We do not further tune the hyperparameters for experiments with VAEs and utilize the same value of $\sigma = 1$ and a decay rate of 0.9 on both datasets.

# D  Initialization Experiments

Table 9: Study on the initialization of the parameters on a ResNeXt-50 [63] for CIFAR-100 [30]. The experiments consider two popular initialization schemes, specifically Kaiming initialization [18] and Xavier initialization [14]. All experiments are run using CBS and we report the mean accuracy over 5 random seeds. The **bolded** number is the setup used for all experiments in Section 4.

| Kaiming init [18] | Xavier Init. [14] |
|---|---|
| **77.0** $\pm$ 0.1 | 76.8 $\pm$ 0.3 |

Here we consider two popular techniques used to initialize parameters in modern deep networks: Kaiming initialization [18] and Xavier initilization [14]. In both of the experiments considered, we see that CBS is insensitive to the initialization scheme used. The networks are able to recover from the initialization are perform comparably on CIFAR-100 using a ResNeXt-50. We use Kaiming initialization [18] for all experiments in the paper, including the baselines.