[Reviews · NeurIPS 2020]

Review 1

Summary and Contributions: Update: The paper has several positives, as noted in all the reviews, but one of my initial concerns was the lack of strong comparisons. The paper mentions several works that can be potentially compared with. The initial reviews also suggested additional methods for comparison. I find the author response on this front partially convincing. The proposed approach can be seen as being orthogonal to some of these, and the comparison shown in the author response is worth including in the main paper. This is a step in the right direction that the final version of the paper, if accepted, should build upon. A discussion wrt [R4] will be another useful addition to the paper. In summary, I find the paper interesting and am leaning towards an accept. This paper presents a way to mitigate the impact of noisy features can have when training a CNN, especially during the initial stages of training. This is achieved by proposing a curriculum based approach that smoothes the features with a Gaussian kernel. The kernel moderates the amount of high frequency information that is propagated during the initial stages of training a CNN. This idea takes inspiration from recent work [21], which proposes a curriculum for learning GANs. The approach presented in the paper is evaluated on several tasks, including: transfer learning, generative models, image classification, feature learning, etc.

Strengths: * The paper presents an effective approach clearly. The ideas are simple, which can potentially make them popular. * The approach is evaluated empirically on several tasks, showing that it is applicable in a varied scenarios. * The details in the paper and the source code that is provided along with submission are sufficient information for reproducibility.

Weaknesses: While the paper presents an effective idea, there are a few questions that need to be addressed. * The choice of using a single kernel size for all the layers of the network (which is changed over the training iterations) seems a bit arbitrary. Would a layer-dependent kernel size not be more appropriate? This choice could also be potentially influenced by the architecture used as well. * The paper mentions several works that have attempted to improve the issues in learning a CNN, but none of them are compared in the empirical evaluation. This is the biggest issue with this paper, as it falls into the category of papers that require a very strong empirical evaluation. This is essential for at least one of the tasks considered in the paper.

Correctness: Yes

Clarity: Yes

Relation to Prior Work: Yes

Reproducibility: Yes

Additional Feedback: There are a few minor typos in the paper.


Review 2

Summary and Contributions: Convolutional neural networks have met tremendous success in many tasks of imaging and signal processing. The paper proposes a new training setup, in which only low frequency information is propagated through the network in the earlier stages of optimization. This is achieved by a smoothing of the CNN features in each layer by a Gaussian kernel. Along training, the width of the kernel is diminished so that higher frequency information is allowed to go through the network. The increased stability of the learning process is particularly useful in the context of GANs.

Strengths: The idea seems elegant; it is difficult for me to evaluate the relevance of the empirical validation. No theoretical result is given.

Weaknesses: The authors claim that no trainable parameter is added, which is true, however the initial choice of sigma and its decrease rate are not theoretically studied; this would make the contribution less heuristic. The impact of curriculum by smoothing is compared to somehow "vanilla" training (sgd with the parameters of the original paper). However there exist many techniques beyond SGD to improve training (eg, as mentioned by the authors, batch normalization), against which the authors do not benchmark.

Correctness: Empirical methodology is limited

Clarity: Yes

Relation to Prior Work: Yes

Reproducibility: Yes

Additional Feedback: Typos: backpropogation (multiple times) a Gaussian Kernel layers perforn Adaptpation (these typos can easily be avoided by using a spellchecker on your test editor)


Review 3

Summary and Contributions: The paper presents a training approach for CNN based on a progressive smoothing of the network by gaussian kernels. The idea is clearly presented and numerous experiments clearly illustrate the approach.

Strengths: The proposed approach is simple, clearly presented, can easily be added to any architecture. No hyperparameter tuning is added, which is a clear benefit of the approach. The experiments are well motivated and illustrate clearly the approach. The idea is well motivated, and quite interesting. Numerous appears considered some kinds of smoothing for deep neural architectures. This paper uses a smoothing that is adapted to the nature of the data (convolutions for images or sounds). The code is well-written and seems easy to use.

Weaknesses: Future directions are possible, e.g. analyzing other smoothing and compare the results to understand the choice of the Gaussian kernel. However the present paper clearly does what it searched for.

Correctness: The experiments present clear comparisons with error bars

Clarity: The paper is well motivated and well-written.

Relation to Prior Work: Yes.

Reproducibility: Yes

Additional Feedback: What is the behavior of the optimization process? Does the smoothing accelerate the optimization ? or hurt it ? Minor comments: Rather than insisting on the elegance of the approach (which is normally left to the reader's judgment), the authors may insist on the fact that their method does not add any additional parameters to tune which is very important advantage. l. 248, 307 regiment -> regime ?


Review 4

Summary and Contributions: The authors present an approach for training convolutional neural networks using Gaussian kernels, which are gradually modified from a high-level of blur to a low-level of blur, simulating a kind of curriculum learning. The authors present experiments on multiple tasks, showing improvements over the standard convolutional neural networks.

Strengths: + The authors present superior results compared to the standard training approach on various tasks, ranging from detection and segmentation to zero-shot domain adaptation and image generation.

Weaknesses: - The authors compared their method to the baseline approach only. However, there are plenty of curriculum learning methods that could have been used as relevant state-of-the-art competing methods to compare with, e.g. [R1, R2, R3, R4]. Comparison with such competing methods is mandatory, in my opinion. - In Eq. (2.1), I believe that the non-linearity is typically applied before the pooling operation. - In terms of novelty, the idea of adding some Gaussian kernels to the network is quite straightforward and simple. Even so, it is not clear why it works so well. The provided motivation is not enough. I would have like to see some visualizations of low-level, mid-level and high-level filters and how these evolve during training in order to figure out what is happening. All the experiments are performed on images, so I would consider this a vision paper. A vision paper without figures is not a properly written vision paper. - Does the approach apply to data other than images? Until proven otherwise, it should clearly stated in the title that the approach applies to images only, e.g. "Curriculum by Smoothing for Images". - Are the improvements statistically significant? A statistical test should be performed to test the null hypothesis. Missing references: [R1] Saxena, S., Tuzel, O. and DeCoste, D., 2019. Data parameters: A new family of parameters for learning a differentiable curriculum. In Advances in Neural Information Processing Systems (pp. 11093–11103). [R2] Soviany, P., Ardei, C., Ionescu, R.T. and Leordeanu, M., 2020. Image difficulty curriculum for generative adversarial networks (CuGAN). In The IEEE Winter Conference on Applications of Computer Vision (pp. 3463-3472). [R3] Penha, G. and Hauff, C., 2020, April. Curriculum Learning Strategies for IR. In European Conference on Information Retrieval (pp. 699-713). [R4] Karras, Tero, Timo Aila, Samuli Laine, and Jaakko Lehtinen. "Progressive growing of gans for improved quality, stability, and variation." arXiv preprint arXiv:1710.10196 (2017).

Correctness: Seems correct, but a comparison with competing methods is missing.

Clarity: There are some typos and English mistakes to be corrected: - "Gaussian kernels are a deterministic functions" => "Gaussian kernels are deterministic functions"; - ". n the sample pseudo-code" => ". In the sample pseudo-code"; - "[31, 30, 45, 8]" => "[8, 30, 31, 45]" (references should be provided in order).

Relation to Prior Work: There are some missing references, e.g. [R1, R2, R3, R4].

Reproducibility: Yes

Additional Feedback: The main drawback is that the authors have only compared with the standard CNN training. Hence, it is unclear how the method compares to other curriculum learning methods. Another important issue is the lack of visualizations: it is not clear how and why the method works. I have the following observations regarding authors' response: 1. Regarding the requirement to compare with competing methods, the authors mentioned that related works based on curriculum are orthogonal. However, there are works that are not orthogonal. For example, [R4] is a method that applies curriculum using a very similar idea. In [R4], the authors progressively increase the size of input, starting with low-resolution images and increasing their size until they are able to generate realistic high-resolution images. I believe that the idea of smoothing the kernels is very similar to [R4], i.e. applying smoothed kernels on large images is equivalent to applying sharp (non-smoothed) kernels on smaller images. Since there are non-orthogonal approaches, e.g. [R4], I believe that a comparison with other curriculum learning methods is still mandatory. 2. The authors did not address my comment regarding visualization of kernels during training. I believe it is important to see how the kernels from some layer converge with and without smoothing. It could explain why and when the proposed idea is useful. Could be inserted in the supplementary at least. 3. It is not a problem that the authors present results on a single data modality: images. However, the contribution should be stated accordingly. For example, I am not sure that smoothing kernels applied on text data would have the same effect. [R4] Karras, Tero, Timo Aila, Samuli Laine, and Jaakko Lehtinen. "Progressive growing of gans for improved quality, stability, and variation." arXiv preprint arXiv:1710.10196 (2017).

[Author Response · NeurIPS 2020]

We thank each of the reviewers for their time to review the paper and for providing feedback on the submission. We
provide a brief summary of the reviews, and detail each reviewers concerns regarding the submission individually: The
submission is well-written [R1] ,[R2] ,[R3] and proposes a simple and effective idea [R1] , [R2] , [R3] which can be
applied in a diverse setting to improve learning in CNNs [R1] , [R3] . The idea is well motivated [R3] and elegant [R2] ,
and outperforms the baselines in diverse contexts [R1] , [R3] such as image classification, feature learning, generative
models, among others [R1] , [R3] , [R4] . The experimental results are convincing across the different settings [R1] ,
[R3] . The code submitted is simple, and easy to follow [R1] , [R3] which aids reproducibility [R1] , [R2] , [R3] , [R4] .

**[R1] : The choice of using a single kernel size for all the layers of the network.** We provide a discussion over the
effect of adding CBS to single layer of the network in Appendix A, but one of the main benefits of using CBS is that the
method works well without extensive hyperparameter tuning. It may be possible that the empirical performance can be
further improved by adjusting the kernel-size and for the different layers in the network, but simply using a kernel-size
of 3 strongly suggests the robustness of the proposed solution.

**[R2] + [R1] : there exist many techniques beyond SGD to improve training (eg, as mentioned by the authors,**
**batch normalization), against which the authors do not benchmark.** The main benefit of the paper is that **CBS**
**augments the already existent CNN improvements, such as batch normalization and dropout**. More specifically,
techniques like batch normalization already exist in ResNet, WideResNet and ResNeXt architectures; similarly VGG
network utilize Dropout. Since each of the network architectures are used as proposed, **this strengthens the empirical**
**evaluation since CBS improves models without changing the underlying CNN improvements that already exist**.

**[R2] : choice of $\sigma$ and its decrease rate are not theoretically studied:** We provide a discussion over the choice of $\sigma$
and the decay rate in the Appendix C, empirically. We will elaborate on the choices in the paper for the final draft.

**[R3] : analyzing other smoothing and compare the results to understand the choice of the Gaussian kernel.** We
analyze the effect of different smoothing curricula in Table 7. The different common kernels that exist are linear kernels,
box-kernels (similar to average-pooling), Laplacian kernel, among others, which do not provide smoothing. There
may be kernels of different forms that have similar behaviour in practice, but a Gaussian kernel is effective and easy to
implement, and has been studied thoroughly in signal and image processing.

**[R3] : Does the smoothing accelerate the optimization?** We perform additional experiments and monitor the epoch
number when the best validation accuracy is reached for WideResNet trained on CIFAR-100. For both the baseline and
CBS the best validation accuracy is reached at epoch $90.8 \pm 0.6$ and $91.0 \pm 0.8$, which suggests that **the optimization**
**is similar to the baseline, while achieving better performance.** Similar results are seen for other architectures.

**[R4] : there are plenty of curriculum learning methods that could have been used as relevant state-of-the-art**
**competing methods to compare with.** Curriculum learning papers work by adjusting the order of sampling or using
importance weights, which is orthogonal to CBS, since the batch sampling modifications can be added on top of CBS
as well. But for completeness and given the short rebuttal period, we run the comparison against [R2] in the suggested
references, since its closely related to [R3] as well on the CIFAR-10 dataset using importance weighting based on image
difficulty, as proposed in [R2]. **For both experiments the suggested baseline does not improve vanilla CNNs**: 91.7
$\pm$ 0.3 and 92.7 $\pm$ 0.2, for Wide-ResNet and ResNeXt, respectively. One possible reason for why weighting-based
methods do not outperform a vanilla CNN, in supervised learning, may be due to overparameterization of NNs resulting
in memorization of all data as suggested in [1] [1]. CBS does not depend on importance weights or data-sampling. CBS
is also inherently simpler than typical curriculum learning methods, as it does not rely on a measure of "task-difficulty"
compared to the citations suggested by Reviewer 4 [R2, R3]. We will add the results and discussion to the final draft.

**[R4] : non-linearity is typically applied before pooling** We will fix the equation as suggested in the final draft.

**[R4] : Does the approach apply to data other than images?** We study the effect of training CNNs with a Gaussian
filter since images, unlike text or tabular data, have strong relations to signal processing. The inherent simplicity of the
technique make it possible such that similar benefits can be gained on non-image data, but typically it is sufficient for a
paper to show experiments with only data of one modality. The diverse nature of experiments does suggest that CBS
improves CNNs in a variety of different contexts beyond supervised learning.

**[R4] : Are the improvements statistically significant?** In machine learning literature, the standard of reporting
the mean and standard deviation with 5 random seeds is performed for all experiments. We do note that for most
experiments, the sum of the baseline and its standard deviation is lower than the score for CBS minus its standard
deviation, which suggests that the results are indeed statistically significant.

**[R1] [R2] [R3] [R4] Missing citations and typos** Thank you for the suggestions; we have made the suggested
corrections.

## Footnotes

[1][1] Byrd, Jonathon, and Zachary Lipton. "What is the effect of importance weighting in deep learning?." ICML. 2019.


[Meta-Review · NeurIPS 2020]

The reviews of this paper are positive overall. This paper presents a curriculum based approach that smoothes the features with a Gaussian kernel in order to mitigate the impact of noisy features can have when training a CNN, especially during the initial stages of training. The proposed approach is evaluated on several tasks: transfer learning, generative models, image classification, representation learning, etc. The reviewers appreciated the simplicity and the effectiveness of the proposed approach. A reviewer comments that the proposed approach is 'simple, clearly presented, can easily be added to any architecture' and that 'no hyperparameter tuning is added, which is a clear benefit'. A reviewer observed that 'the code is well-written' and 'easy to use'. The authors submitted a response to the reviewers' comments. After reading the response, updating the reviews, and discussion, the reviewers feel that the authors provide 'extensive results', present 'an ablation study to understand their approach', and that 'the paper has sufficient merit'. The AC decision making was based on constructive comments only. We recommend to take the reviewers' comments and suggestions into account while preparing the final version of the paper. Recommendation for spotlight. Curriculum learning is a highly popular topic in machine learning and beyond. Yet, as of today, before this paper, no method seemed to stand out as both simple and practical. This paper proposes such a method. The proposed method has a high potential for a broad impact because of its striking mathematical and numerical simplicity and its excellent performance on several tasks. Indeed, as R1 says, 'the approach is evaluated empirically on several tasks, showing that it is applicable in a varied scenarios’. Moreover, as R1 and R3 say, ‘the code is well-written and seems easy to use’. All in all, this paper makes an important contribution to the field, with a (finally) simple and clear approach for curriculum learning. Accept.